# New Inhibitors of the Human p300/CBP Acetyltransferase Are Selectively Active against the Arabidopsis HAC Proteins

**DOI:** 10.3390/ijms231810446

**Published:** 2022-09-09

**Authors:** Chiara Longo, Andrea Lepri, Andrea Paciolla, Antonella Messore, Daniela De Vita, Maria Carmela Bonaccorsi di Patti, Matteo Amadei, Valentina Noemi Madia, Davide Ialongo, Roberto Di Santo, Roberta Costi, Paola Vittorioso

**Affiliations:** 1Department of Biology and Biotechnology “Charles Darwin”, Sapienza University of Rome, 00185 Rome, Italy; 2Department of Chemistry and Technology of Drug, Sapienza University of Rome, Istituto Pasteur Italia-Fondazione Cenci Bolognetti, 00185 Rome, Italy; 3Department of Environmental Biology, Sapienza University of Rome, 00185 Rome, Italy; 4Department of Biochemical Sciences, Sapienza University of Rome, 00185 Rome, Italy

**Keywords:** *Arabidopsis thaliana*, HAC proteins, p300/CBP inhibitors

## Abstract

Histone acetyltransferases (HATs) are involved in the epigenetic positive control of gene expression in eukaryotes. CREB-binding proteins (CBP)/p300, a subfamily of highly conserved HATs, have been shown to function as acetylases on both histones and non-histone proteins. In the model plant *Arabidopsis thaliana* among the five CBP/p300 HATs, HAC1, HAC5 and HAC12 have been shown to be involved in the ethylene signaling pathway. In addition, HAC1 and HAC5 interact and cooperate with the Mediator complex, as in humans. Therefore, it is potentially difficult to discriminate the effect on plant development of the enzymatic activity with respect to their Mediator-related function. Taking advantage of the homology of the human HAC catalytic domain with that of the Arabidopsis, we set-up a phenotypic assay based on the hypocotyl length of Arabidopsis dark-grown seedlings to evaluate the effects of a compound previously described as human p300/CBP inhibitor, and to screen previously described cinnamoyl derivatives as well as newly synthesized analogues. We selected the most effective compounds, and we demonstrated their efficacy at phenotypic and molecular level. The in vitro inhibition of the enzymatic activity proved the specificity of the inhibitor on the catalytic domain of HAC1, thus substantiating this strategy as a useful tool in plant epigenetic studies.

## 1. Introduction

Histone acetylation is one of the main post-translational modifications (PTMs) involved in the epigenetic control of gene expression, mainly in eukaryotes. It occurs through the transfer of the acetyl group of the acetyl-CoA donor substrate, onto the amino groups of the lysine residues of the histone N-terminal tails, by specific proteins or complexes [1]. Histone acetyltransferases (HATs) and deacetylases (HDACs) are among the best structurally characterized histone modification enzymes. The activity of HATs and HDACs is related to gene activation and repression, respectively, since acetylation relaxes chromatin by neutralizing the net positive charge of lysine interacting with the phosphate backbone of DNA. These enzymes play fundamental roles in a large number of developmental processes and their deregulation has been linked to the progression of diverse human disorders, including cancer [2,3,4]. Interestingly, several of these enzymes have been shown to also acetylate non-histone proteins [5,6] such as the p53 tumor suppressor [7] and the E1A viral oncoprotein [8].

CREB-binding proteins (CBP)/p300, a subfamily of highly conserved HATs, have been shown to function as acetylases on both histones and non-histone proteins, but also as co-activators or scaffold proteins in regulatory complexes [9]. The human CBP/p300 KAT3 (lysine acetyltransferase type 3) acetyltransferases is known to interact with at least 400 proteins, playing a central role in several regulatory networks [10]. Previous studies revealed a tight cooperation of CBP/p300 with the Mediator transcription complex in mammals [11,12].

In plants there are four sub-families of HATs: HAG, related to the human GNAT (GCN5-related N-terminal acetyltransferases); HAM, related to the MYST family; HAF, or HATs of the TAFII250 family; and HAC, acetyltransferases of the CBP/p300 family [13]. The Arabidopsis genome encodes five CBP/p300 HAT proteins (HAC1, HAC2, HAC4, HAC5 and HAC12) [14]. Among the Arabidopsis HAC proteins, HAC2 was shown not to have in vitro acetylase activity [15], consistent with the early divergence suggested by the phylogenetic analysis [14]; similarly, HAC4 also lacks acetylase activity, and it has been proposed to be an expressed pseudogene [16]. On the other hand, the catalytic domain of the Arabidopsis HAC1 has been shown to be conserved with the human p300/CBP, and to display in vitro acetylase activity, similarly to HAC5 and HAC12 [15,16].

In agreement with the involvement of HACs in several plant developmental processes, inactivation of the *HAC1* gene has been shown to affect both the *Arabidopsis* vegetative and reproductive development; indeed, *hac1* mutations lead to a late-flowering phenotype and pleiotropic developmental defects, such as short primary roots and reduced fertility [17,18,19]. Interestingly, combination of *hac* mutants, namely *hac1*, *hac5*, and *hac12*, displays pleiotropic phenotypes associated with hypersensitivity to ethylene in both dark and light conditions. Consistently, the *hac1hac5* double mutant shows increased expression of the *ETHYLENE RESPONSE FACTOR* (*ERF*) genes *ERF1*, *ERF4*, *ERF6* and *ERF11*, thus suggesting that plant HACs are involved in the ethylene signaling pathway [20]. A more recent study, focused on leaf senescence, has shown that *hac1* mutant alleles display delayed age-related leaf senescence. The Authors also identified several potential HAC1 targets, with ERF022 as a positive regulator of this phenomenon [21]. In the positive regulation of *ERF022*, HAC1 acts additively with MED25, a subunit of the Mediator complex, previously shown to interact with HAC1, and to regulate transcription during jasmonic acid signaling [22]. More recently it has been shown that both HAC1 and HAC5 are recruited by the subunits MED8 and 25 to the Mediator complex, and these HAC and MED factors cooperate in the control of the repressor of flowering, FLOWERING LOCUS C (FLC), through the positive regulation of the flowering-inducing gene *GLYCINE-RICH RNA-BINDING PROTEIN*
*7* (*GRP7*) [23].

Although not widely used in plant research, the use of epidrugs has proved to be an effective approach in the study of epigenetic modifications in Arabidopsis. Besides the widely used HDAC inhibitor Trichostatin A (TSA) [24], the HDAC inhibitor Ky-2, previously tested in human cells [25,26], has been used to deepen the study of HAC/HDAC in the response to salt stress in Arabidopsis [27]. Ky-2 treatment increased total histone acetylation level and resulted in enhanced expression of salt-responsive genes and consequent increased stress tolerance [27]. As for chemical inhibitors of HATs in plants, the efficacy of the compound MB-3, previously designed and tested against human GCN5 (General Control Non-repressed 5 protein) [28], has been evaluated in Arabidopsis seedlings [29]. The Authors showed that MB-3 caused a decrease in total histone acetylation level, as well as a reduction in the transcript level of direct GCN5 target genes. These molecular effects were associated, at physiological level, with shorter roots and chlorotic leaves [29]. More recently, we have assessed the efficacy and selectivity of an inhibitor of the human catalytic subunit of Polycomb Repressive Complex2 (PRC2), responsible for the trimethylation of lysine 27 on histone H3, on Arabidopsis seeds and seedlings [30]. Indeed, treatment with the compound RDS 3434, resulted in a significant decrease in H3K27me3 and of the expression level of known PRC2 target genes, affecting both seed germination and root growth [30].

Given the presence of HAC1 and HAC5 in the Mediator complex, it sounds tricky to discern between HAC function as part of the Mediator Complex and its acetylase activity. To inhibit the acetylase, using an epidrug will provide a tool to distinguish the effect on plant development of the enzymatic activity with respect to its Mediator-related function.

Taking advantage of the homology of the human HAC catalytic domain with that of the Arabidopsis, we set-up a phenotypic assay based on the hypocotyl length of dark-grown seedlings to evaluate the effects on Arabidopsis seedlings using compound **1a**, previously described as human p300/CBP inhibitor [31]. We then screened cinnamoyl derivatives **1b**–**f**, structurally related to **1a** and previously described as inhibitors of human p300/CBP [32], and their newly synthesized analogues **2a**–**c** (Figure 1).

## 2. Results

### 2.1. Synthesis of HAC Inhibitors

The tested compounds **1a**–**f** were synthesized as previously described [31,32]. The synthesis of compounds **2a**–**c** is outlined in Figure 2. The 2-bromo-4-(bromomethyl)phenol [33] underwent a SN_2_ reaction with the *N*,*N′*-trimethyleneurea in the presence of NaH (60% dispersion in mineral oil) to obtain derivative **2a**. Similarly, 2-bromo-4-(bromomethyl)phenol reacted with the enamine 1-(1-cyclohexen-1-yl)pyrrolidine [34] to give the iminium salt that was subsequently hydrolyzed back in acidic medium into the carbonyl, giving the cyclohexanone derivative **3**. Lastly, a demethylation reaction with BBr_3_ furnished the phenol derivatives **2b** and **2c**. The detailed synthetic procedures, the analytical and spectroscopic data of the synthesized compounds are reported in Section 4.

### 2.2. Evolutive Conservation Analysis of the HAC1 Catalytic Site

Sequence alignment of the catalytic subunit of the Arabidopsis HAC1 protein with the human p300/CBP have previously revealed 35% identity within this functional domain [15]. We have performed an additional study, using the ConSurf server ([35], https://consurf.tau.ac.il; accessed on 26 December 2021), to evaluate the conservation, from an evolutionary point of view, of the amino acid residues of the catalytic domain (amino acids 1075–1511), based on the phylogenetic relationships between homologous sequences. Indeed, the degree of evolutionary conservation of an amino acid position strongly depends on its structural and functional value. Thus, the analysis of the conservation of each amino acid among members of the same protein family can reveal the importance of each position for the structure or function of the protein itself. ConSurf allows the accurate evaluation of the evolutionary rate, obtained using an empirical Bayesian method that considers the stochastic process underlying evolution, multiple sequence alignment and the basic phylogenetic tree. In addition, the conservation levels are normalized and grouped into nine degrees, where 1 indicates the positions with a faster evolution, and 9 indicates the most conserved positions in evolutionary terms. The degrees of conservation are then mapped to the query amino acid sequence using a cyan-violet color code, for less and more conserved positions, respectively. As shown in Figure 3, the most important residues from a functional and structural point of view, and consistently the most conserved, are those between positions 1195–1206, 1255–1274 and 1485–1511. Interestingly, these are the same residues with which the inhibitor comes into contact, according to the analysis by Madia et al. (2017) [32].

### 2.3. The p300/CBP Inhibitor ***1a*** Affects Hypocotyl Elongation in Arabidopsis Seedlings

The inhibitory effect of the compound **1a** was previously evaluated in HeLa H2B-EYFP cells, where a 50% reduction in H3 acetylation was attained with a 40 μM concentration. The inhibitor **1a** was also proven to be highly selective for the p300/CBP HATs, compared to other classes of HATs, namely GCN5 and PCAF (p300/CBP Associated Factor) [31,32]. Taking advantage of the homology of the human HAC catalytic subunit with that of the Arabidopsis, we have assessed the efficacy of **1a** in plants. Given that inactivation of *HAC1*, in combination with other *hac* mutations, resulted in ethylene hypersensitivity with a reduction in hypocotyl elongation [20], we set-up a phenotypic assay based on the hypocotyl length of dark-grown seedlings, since the hypocotyls of etiolated plants allow a more reliable measurement than de-etiolated seedlings. We measured the hypocotyl length of 6-day-old seedlings. Wild type seedlings were grown in darkness for 2 days, then transferred on a medium supplied with **1a** (60, 120 μM), or with DMSO as a control, for additional 4 days. Interestingly, treatment with **1a** 120 μM resulted in a significant reduction in the hypocotyl length, with a dose–response trend, compared to the DMSO-treated control (Figure 4A).

### 2.4. Screening and Selection of New HAC Inhibitors

Once the experimental set-up was defined and based on the promising results on the efficacy of **1a** on Arabidopsis seedlings, we performed a screening on compounds **1b**–**f**, previously described as inhibitors of human p300/CBP, and their newly synthesized analogues **2a**–**c**. These compounds were analyzed in a phenotypic assay on the hypocotyl length. The results clearly showed that **1b**, **2b** and **2c** were the most effective compounds, as the treatment with these compounds caused a statistically significant reduction in the hypocotyl length. Compounds **1b** and **2c** affect the hypocotyl length only at the highest concentration while, curiously, **2b** does not show an increased reduction in the hypocotyl length at the highest concentration (120 μM) (Figure 4B).

Based on these results, we reasoned that, regardless of the concentration supplied to the medium, the amount of compound taken up by the seedlings could be limiting. Therefore, a screening with higher concentrations (120 and 240 μM) was carried out, with the selected compounds **1b**, **2b** and **2c** as hit candidates compared to **1a**. The results of this phenotypic analysis revealed that the **2c** inhibitor was the most effective compound, with a statistically significant reduction in the hypocotyl length at both concentrations. Once again, **2b** caused a statistically significant reduction in the hypocotyl length at the lowest concentration, with a peculiar slight opposite effect at 240 μM, with longer hypocotyls, albeit to a lesser extent compared to the control. Treatment with **1b** resulted in a slight, although not significant, reduction in the hypocotyl length, similarly to compound **1a** (Figure 4C,D). Taken together, these phenotypic assays proved that **1b**, **2b** and **2c** are the most promising inhibitors, therefore they have been chosen for subsequent analysis, in comparison with **1a**.

### 2.5. Treatment with Compounds ***2b*** and ***2c*** Mimics Ethylene Effect on hac Mutants

The HAC acetyltransferases have been shown to be involved in the ethylene signaling pathway, thus we wondered whether the effect of the inhibitors on the hypocotyl length would be enhanced by a combined treatment of inhibitors and the precursor of ethylene 1-aminocyclopropane-1-carboxylate (ACC) (10 μM). As ethylene in the dark represses hypocotyl elongation [36,37], the addition of the ethylene precursor resulted in shorter hypocotyls in wild type dark-grown seedlings (MOCK + ACC). Compared to the ACC-treated control and to the 18% reduction in the hit compound **1a** (120 μM), treatment with the selected inhibitors added with ACC caused a sharper phenotype with a significant reduction in the hypocotyl length. Indeed, **2b** and **2c** produced a significant decrease in hypocotyl elongation at both concentrations (38% and 39% **2b**, and 31% and 26% **2c** at 120 and 240 μM, respectively), whereas only treatment with 240 μM **1b** resulted in a significantly reduced hypocotyl length (31%) (Figure 5A). These results prove that the experimental set-up is sound and allowed us to identify **2b** and **2c** as the most effective compounds on Arabidopsis seedlings.

Since the addition of ACC to the medium in the presence of the selected inhibitors sharpened the phenotype, further decreasing hypocotyl length, we wondered whether this phenotype was comparable to that of the single *hac* mutants, namely *hac1* and *hac5*. To this end, we measured the hypocotyl length of dark-grown single mutants *hac1* and *hac5* (SALK_080380 and SALK_074472, respectively), and of the hypocotyls of **2b**- or **2c**-treated seedlings. Since a shorter hypocotyl phenotype has been shown only for the *hac1hac5* double mutant [20], and given the involvement of HAC family proteins with the ethylene signaling pathway, the phenotypic analysis was also carried out in the presence of ACC. As expected, the hypocotyl length of the *hac1* and *hac5* single mutants was similar to that of the wild type, whereas, in the presence of ACC, the single mutants showed a slightly greater, although not significant, reduction in hypocotyl elongation compared to the ACC-treated control. Interestingly, hypocotyls of **2c**-treated seedlings were significantly shorter than the *hac1* and *hac5* single mutants in the presence of ACC, showing a 26% and 27% reduction, respectively, at 120 and 240 μM. Intriguingly, treatment with **2b**, with the addition of ACC, gave rise to a significant decrease (18%) only at the lowest concentration, with the same trend observed in the absence of ACC (Figure 5B,C). This evidence suggests that these hit inhibitors are specifically active on acetyltransferases, rather than acting in an off-target manner, mimicking and even emphasizing the hypocotyl phenotype of *hac* single mutants.

### 2.6. Treatment with ***2b*** and ***2c*** Affects Expression of HAC Target Genes

In Arabidopsis, it has been shown that HAC1 and HAC5 are subunits of the Mediator complex [23]. To identify HAC1 and HAC5 target genes, we used RNA-Seq and ChIP-Seq data from Guo et al. (2021) [23]. In this study, the *hac1hac5* transcriptome was compared with the one of the double mutants of the Mediator subunits *med8med25*. In addition, they also performed a H3ac ChIP-seq of both *hac1hac5* and *med8med25* double mutants. We focused our interest on the *hac1hac5*-specific downregulated genes, which we compared with the genes with reduced H3Ac peaks in *hac1hac5* [23]. From this comparative analysis, we have selected four target genes: *HLS1* (*HOOKLESS1*), an ethylene-response gene encoding a putative N-acetyltransferase involved in the auxin signaling pathway [38], *ORG1* (*OBP3 RESPONSIVE GENE1*), which encodes a chloroplast-localized protein involved in stress response [39], *LTP5* (*LIPID TRANSFER PROTEIN 5*) [40], whose protein is involved in lipid transfer and *LSH6* (*LIGHT*-*DEPENDENT SHORT HYPOCOTYLS 6*) [41], which is involved in light-mediated seedling development. The expression analysis was performed on wild type seedlings grown in darkness for 4 days, then transferred on a medium supplied with **2b** or **2c** (120, 240 μM), or with DMSO as a control (MOCK), for additional 2 days. Treatment with **2c** resulted in a significant decreased expression of the four target genes, indicating that this compound is clearly effective on Arabidopsis seedlings (Figure 6A). Consistently, with the effect on the hypocotyl length, **2b** treatment gave opposite results at 120 and 240 μM concentrations, as it resulted in a significantly reduced expression of *HLS1*, *ORG1*, *LTP5* and *LSH6* at the lowest concentration, while causing a significant increase at 240 μM (Figure 6A).

It has been previously shown that expression of the *ETHYLENE RESPONSE FACTOR 1* (*ERF1*), *ERF4*, *ERF6* and *ERF11* genes is increased in the *hac1hac5* double mutant [20], consistent with the ethylene hypersensitive phenotype. As treatment with **2b** and **2c** strengthens the effect of ethylene on the hypocotyl length, similarly to the lack of both HAC1 and HAC5, we wondered whether, in the presence of compounds **2b** and **2c**, expression of *ERF* genes could be increased. Therefore, we assessed the transcript level of *ERF4* and *ERF6* in **2b** and **2c**-treated *hac1* seedlings, compared to the expression in the *hac1* single mutant and in the mock-treated wild type control. According to the previous data presented by Li et al. (2014) [20], the lack of HAC1 results in a slight but significant increase in both *ERF4* and *ERF6* expression compared to the control (Figure 6B). Interestingly, in the **2b** and **2c**-treated sample, *ERF6* expression was sharply and significantly increased in a dose-dependent manner with respect to both the untreated *hac1* mutant and wild type control, while *ERF4* transcript level was enhanced significantly but only with the highest concentration (Figure 6B), thus corroborating the ethylene-related phenotypic data.

### 2.7. Expression of HAC1 and HAC5 Is Mutually Regulated

The HAC1 and HAC5 proteins are likely to have at least partially redundant functions, as the single *hac1* and *hac5* mutants do not display striking phenotypes at the seedling stage, while the double mutant shows the most severe phenotype for hypocotyl and root length, as well as at the adult plant stage [20]. Therefore, we wondered whether HAC1 and HAC5 would mutually affect their expression. To verify this hypothesis, we analyzed the expression of both *HAC1* and *HAC5* in the single *hac5* and *hac1* mutant seedlings, respectively, by RT-qPCR. Interestingly, both *HAC5* and *HAC1* showed a significant increase in mRNA level in the absence of HAC1 and HAC5, respectively (Figure 7). This result suggests that there may be a kind of compensation in the steady-state mRNA levels of both these genes and confirms the partial functional redundancy of these two acetylases [20].

### 2.8. Seedlings Effectively Uptake HAC Inhibitors

Our phenotypic and molecular experiments did not give significant results with a concentration of the inhibitory compounds lower than 120 μM. Although this is not a toxic concentration for plant growth and viability (data not shown), we wondered what amount of the compound was effectively taken up.

Therefore, we designed a specific protocol to measure plant uptake of inhibitor **2c**, since it was the most promising compound. Four-day-old seedlings were transferred on liquid MS medium supplied with compound **2c** (240 μM), or DMSO as a control, for 2 additional days. Seedling uptake was determined indirectly, measuring the residual amount of compound **2c** in the medium. To this aim, samples (S**2c**.1 to S**2c**.3) and controls (C.1 to C.3) solutions were analyzed by HPLC. According to our results, a seedling uptake of 0.25148 mg of compound **2c** was measured, corresponding to a 28.45% of the amount supplied in the medium (Table 1).

### 2.9. Compound ***2c*** Selectively Inhibits HAC1 Activity

The specific activity of the selected inhibitors has not been previously tested on human CBP/p300; this prompted us to assess the efficacy of the most active compound, **2c**, on Arabidopsis HAC1. To this end, we cloned a 1.31 kb-long fragment encoding the catalytic domain (aa 1081–1517) of HAC1 with a C-terminal His tag into the expression vector pET-28a. The recombinant protein was successfully expressed in *E*. *coli* and partially purified by affinity chromatography exploiting the 6XHis tag. The partially purified 52 kDa protein was used for an acetyltransferase activity assay (EpiGentek # P-4003). The results shown in Figure 8 reveal that compound **2c** inhibits acetylase activity of the HAC1 protein. Indeed, treatment with compound **2c** (6 μM) resulted in a reduction in HAT activity of 83% compared to the untreated samples (Figure 8), thus proving both the efficacy and the specificity of this inhibitor.

## 3. Discussion

In this work, we describe the development and application of a pharmacological approach for the study of one of the most widespread epigenetic modifications, histone acetylation, mediated by histone acetyltransferases (HAT), in *Arabidopsis thaliana*.

Histone acetylation is correlated with gene activation; indeed, acetylation loosens the chromatin structure by neutralizing the net positive charge of lysines interacting with the phosphate backbone, and by recruiting reader proteins with a bromodomain, including the same acetyltransferases, which can amplify acetylation through a spreading process.

The HAT subfamily of p300/CBP are widespread in eukaryotes, apart from unicellular eukaryotes such as yeast. Proteomic analyses in mammals have identified about 400 proteins and complexes interacting with the HATs of this subfamily, suggesting a key role of these acetyltransferases as hubs of the complex molecular network involved in several developmental processes [10]. Consistently, lack of p300/CBP proteins results in lethal phenotypes in worms, mice, and flies [42], while mutations in the corresponding genes in humans have been associated with several tumors and diseases [43,44]. Accordingly, in the last decade small molecule inhibitors of CBP/p300 have been developed as promising anticancer agents [32,45,46,47,48].

The homology between the plant and animal p300/CBP proteins is restricted to the C-terminal region, encompassing the HAT catalytic domain and the binding domain of the adenovirus protein E1A [14,15]. In Arabidopsis, despite the presence of five *HAC* genes, the HAT domain is functionally conserved only in HAC1, HAC5 and HAC12 [16]. The characterization of the single Arabidopsis *hac* mutants showed that only inactivation of *HAC1* results in pleiotropic developmental defects: shorter roots, late-flowering phenotype [17], darker green and wrinkly leaves, delayed leaf senescence and protruding gynoecia, and hypersensitivity to the phytohormone ethylene [20,21]. In addition, phenotypic analysis of several combinations of *hac* mutants clearly pointed out that the three active acetyltransferases, HAC1, HAC12, and HAC5, act redundantly, HAC1 with a key role and HAC5 closely cooperating with HAC1 [20]. Despite the high homology of the HAC catalytic domain with the human p300/CBP, and the pleiotropic phenotypes in different developmental processes, the pharmacological approach in plants has not been developed. Taking advantage of the homology of the human HAC catalytic domain with that of the Arabidopsis, we have not only assessed the efficacy on Arabidopsis seedlings of a compound previously described as human p300/CBP inhibitor [31], but also screened newly synthesized analogues and proved the selective efficacy of the most promising analogue, compound **2c**.

p300/CBP activity can be defined as promiscuous, as they acetylate both histone and non-histone proteins, and because they have been shown to function as scaffold proteins of transcriptional complexes [9,10]. Indeed, both in mammals and plants, p300/CBP interact with the Mediator transcriptional complex [11,12,22,23]. Although histone acetylation is supposed to be the most prominent activity of these proteins, which of these functions is crucial for the activation of many target genes is still controversial. Therefore, this pharmacological approach, with inhibitors of the HAT activity, may be useful in applied as well as in basic research.

We have assessed the reliability of the hypocotyl elongation inhibition assays as a phenotypic screening of HAC inhibitors. *hac* mutants show an ethylene hypersensitivity phenotype, which we have mimicked in wild type seedlings treated with inhibitors and ACC. Interestingly, the phenotype of treated wild type samples was more severe than that of *hac1* mutant, indicating this phenotypic screening as a useful tool to evaluate the efficacy of new inhibitory compounds. Indeed, treatment with the most active inhibitors that we have identified, namely compounds **2b** and **2c**, results in both phenotypic and molecular effects with a similar trend. Surprisingly, while treatment with compound **2c** results in a dose–response effect, treatment with the highest concentration of compound **2b** causes an opposite effect mainly at molecular level. Indeed, expression of the target genes was decreased in seedlings treated with the low concentration of **2b**, while it was significantly increased respect to the control at the higher concentration. Consistently, a similar trend was also present, albeit to a lesser extent, in the phenotypic assays. A possible explanation is that compound **2b** is somehow more specific for HAC1 than other HACs, and a reduction in HAC1 activity triggers a compensation mechanism through HAC5, resulting in increased acetylase activity. On the other hand, it has previously been reported that some inhibitory compounds can result in a slight increase in acetylase activity. A possible explanation is that these inhibitors can bind to the HAT enzyme in a different conformation and, consequently, function as agonists instead of antagonists [32]. Our results are consistent with a compensation phenomenon between HAC1 and HAC5. In fact, the *HAC1* and *HAC5* genes are mutually regulated, as shown by the increased expression levels of *HAC1* in the *hac5* mutant and of *HAC5* in the *hac1* mutant. Furthermore, the strong reduction in HAC1 acetyltransferase activity by compound **2c** clearly indicates that this small molecule functions by inhibiting the catalytic domain of HAC1.

The Arabidopsis HAC proteins have been shown to be involved in the ethylene signaling pathway [20]. Consistently, several ethylene-responsive genes were up-regulated in the *hac1hac12* double mutant transcriptome, and slightly induced in the single *hac1* and *hac5* mutants [20]. Consistently, treatment of the *hac1* mutant with compounds **2b** and **2c** results in a significant increase in the expression levels of *ERF4* and *ERF11* compared to both the DMSO-treated mutant and wild type controls, thus corroborating the inhibitory effect of these drugs.

The hit compound **1a** was previously tested at a concentration 50 μM on human leukemia cells to verify its efficacy in inducing cell death [31,32]. Our phenotypic analysis was performed on Arabidopsis seedlings with 120 to 240 μM concentrations. Although our assay is on multicellular organisms rather than cell cultures, the concentrations might seem too high. Interestingly, our uptake experiments, performed with 240 μM of compound **2c**, reveal that the in planta amount corresponds to 0.25148 mg, thus further strengthening the efficacy of this compound.

## 4. Materials and Methods

### 4.1. Chemistry

#### 4.1.1. General Instrumentation

Melting points were determined on a Bobby Stuart Scientific SMP1 melting point apparatus and are uncorrected. Compound purity was always >95% as determined by combustion analysis. Analytical results agreed to within ±0.40% of the theoretical values. IR spectra were recorded on a PerkinElmer (Shelton, CT, USA) Spectrum-One spectrophotometer; ^1^H NMR spectra were recorded at 400 MHz on a Bruker (Billerica, MA, USA) AC 400 Ultrashield 10 spectrophotometer (400 MHz). Dimethyl sulfoxide-*d*_6_ 99.9% (CAS 2206-27-1), deuterochloroform 98.8% (CAS 865-49-6) and acetone-*d*_6_ 99.9% (CAS 666-52-4) of isotopic purity (Aldrich, St. Louis, MO, USA)) were used. Column chromatographies were performed on silica gel (Merck; Darmstadt, Germany; 70−230 mesh). All compounds were routinely checked on TLC by using aluminum-baked silica gel plates (Fluka Honeywell, Charlotte, NC, USA; DC-Alufolien Kieselgel 60 F_254_) or TLC aluminum oxide 60 F_254_ basic (Merck). Developed plates were visualized by UV light. Solvents were reagent grade and, when necessary, were purified and dried by standard methods. Concentration of solutions after reactions and extractions involved the use of rotary evaporator (Büchi; Flawil, Switzerland) operating at a reduced pressure (ca. 20 Torr). Organic solutions were dried over anhydrous sodium sulfate (Merck). All solvents were freshly distilled under nitrogen and stored over molecular sieves for at least 3 h prior to use. Analytical results agreed to within ±0.40% of the theoretical values.

#### 4.1.2. General Experimental Procedure

##### General Procedure A (GP-A) for the Synthesis of Derivatives **2b** and **2c**

To a solution of methoxy derivatives (**2b** and **2c**) (1 mmol) in dichloromethane (30 mL) was cooled at −45 °C and BBr_3_ 1 M in dichloromethane (3.3 mmol) was added dropwise. The mixture was stirred at room temperature for 2 h. Upon the completion of reaction as monitored by thin layer chromatography (TLC), the mixture was poured into cold water (50 mL). For compound **2c**, the mixture was extracted with chloroform (3 × 20 mL) and the collected organic layers were washed with brine (4 × 20 mL), dried over Na_2_SO_4_ and evaporated at reduced pressure. For compound **2b**, the organic phase was evaporated under vacuum and the solid that formed was filtered. The residue was washed with water (3 × 10 mL), diethyl ether (2 × 10 mL) and then dry under IR lamp. The raw product was purified by column chromatography on silica gel. For each derivative eluent system; yield (%); melting point; IR; ^1^H NMR and elemental analysis are reported.

#### 4.1.3. Specific Procedures and Characterization

##### 1,3-Bis(3-bromo-4-methoxybenzyl)tetrahydropyrimidin-2(1*H*)-one (**2a**)

A solution of *N*,*N*′-trimethyleneurea (1.10 g, 11 mmol) in 1,4-dioxane *dry* (50 mL) was added to NaH 60% dispersion in mineral oil (2.96 g, 74 mmol) and the reaction mixture was refluxed for 1 h. Next, 2-Bromo-4-(bromomethyl)-1-methoxybenzene (18.19 g, 65 mmol) was added and the resulting mixture was refluxed overnight. The solvent was evaporated under reduced pressure and the raw was extracted with ethyl acetate (25 mL). The organic layers were collected and were washed with water (3 × 20 mL), dried over anhydrous Na_2_SO_4_, filtered and evaporated. The crude material was purified by column chromatography on silica gel using a mixture of *n*-hexane/ethyl acetate (60:40) to afford a white solid (2.7 g, yield 50%); 189–190 °C. IR ν 2858 (OCH_3_), 1621 (CO) cm^−1^; ^1^H NMR (400 MHz, CDCl_3_) *δ* 1.81 (q, 2H, CH_2_-*CH_2_*-CH_2_), 3.10 (t, 4H, J = 6 Hz, *CH_2_*-CH_2_-*CH_2_*), 3.82 (s, 6H, CH_3_), 4.44 (s, 4H, benzyl CH_2_), 6.79 (d, 2H, J*o* = 8.4 Hz, phenyl C5-H), 7.16 (dd, 2H, J*o* = 8.4 Hz, J*m* = 2 Hz, phenyl C6-H), 7.41 (d, 2H, J*m* = 2 Hz, phenyl C2-H). Anal. Calcd for C_20_H_22_Br_2_N_2_O_3_: C, 48.22; H, 4.45; Br, 32.08; N, 5.62%. Found: C, 48.25; H, 4.43; Br, 32.01; N, 5.59% Appendix A).

##### 1,3-Bis(3-bromo-4-hydroxybenzyl)tetrahydropyrimidin-2(1*H*)-one (**2b**)

Compound **2b** was prepared from **2a** (2.0 g, 4.01 mmol) by means of GP-A. N-hexane/ethyl acetate (75:25); 79% as a white solid; 196–198 °C. IR ν 3640 (OH), 1633 (CO) cm^−1^; ^1^H NMR (400 MHz, DMSO *d_6_*) *δ* 1.79 (q, 2H, CH_2_-*CH_2_*-CH_2_), 3.12 (t, 4H, J = 6 Hz, *CH_2_*-CH_2_-*CH_2_*), 4.35 (s, 4H, benzyl CH_2_), 6.91 (d, 2H, J*o* = 8.4 Hz, phenyl C5-H), 7.07 (dd, 2H, J*o* = 8.4 Hz, J*m* = 2 Hz, phenyl C6-H), 7.37 (d, 2H, J*m* = 2 Hz, phenyl C2-H), 10.13 (bs, 2H, OH). Anal. Calcd for C_18_H_18_Br_2_N_2_O_3_: C, 45.98; H, 3.86; Br, 33.99; N, 5.96%. Found: C, 46.00; H, 3.84; Br, 33.97; N, 6.00% (Appendix A).

##### 2,6-Bis(3-bromo-4-hydroxybenzyl)cyclohexanone (**2c**)

Compound **2c** was prepared from 3 (0.30 g, 0.604 mmol) by means of GP-A. *N*-hexane/ethyl acetate (75:25); 165–167 °C. IR ν 3360 (OH), 1683 (CO) cm^−1^; ^1^H NMR (400 MHz, DMSO *d_6_*) *δ* 1.16–1.29 (m, 2H, cyclohexanone C3-H and C5-H), 1.56–1.70 (m, 2H, cyclohexanone C4-H), 1.87–1.91 (m, 2H, cyclohexanone C3-H’ and C5-H’), 2.21–2.26 (m, 2H, benzyl *CH_a_*H and *CH_a_’*H), 2.61–2.70 (m, 2H, CH), 2.85–2.92 (m, 2H, benzyl CH*H_b_* and CH*H_b_’*), 6.82 (d, 2H, J*o* = 8.4 Hz, phenyl C5-H), 6.95 (dd, 2H, J*o* = 8.4 Hz, J*m* = 2 Hz, phenyl C6-H), 7.27 (d, 2H, J*m* = 2 Hz, phenyl C2-H). Anal. Calcd for C_20_H_20_Br_2_O_3_: C, 51.31; H, 4.31; Br, 34.13%. Found: C, 51.33; H, 4.29; Br, 34.15% (Appendix A).

##### 2,6-Bis(3-bromo-4-methoxybenzyl)cyclohexanone (**3**)

To a solution of DIPEA (1.5 mL, 8.7 mmol) and 2-bromo-4-(bromomethyl)-1-methoxybenzene (2.4 g, 8.7 mmol) in chloroform (25 mL), the 1-(1-cyclohexen-1-yl)pyrrolidine (0.55 g, 3.63 mmol) was added dropwise and the mixture was refluxed for 12 h. Then, HCl 1 N (9.96 mL) was added, and the reaction was stirred at same temperature for further 4 h. The mixture was then evaporated, and the resulting aqueous layer was extracted with ethyl acetate (3 × 20 mL). The organic layers were collected, washed with brine (2 × 20 mL), dried over Na_2_SO_4_ and evaporated at reduced pressure. The crude material was purified by column chromatography on silica gel using *n*-hexane/ethyl acetate (80:20) as eluent to afford a white solid (0.93 g, yield 16%); 146–148 °C. IR ν 2845 (OCH_3_), 1693 (CO) cm^−1^; ^1^H NMR (400 MHz, CDCl_3_) *δ* 1.20–1.29 (m, 2H, cyclohexanone C3-H and C5-H), 1.51–1.55 (m, 2H, cyclohexanone C4-H), 1.66–1.70 (m, 2H, cyclohexanone C3-H’ and C5-H’), 2.24–2.30 (m, 2H, benzyl *CH_a_*H and *CH_a_’*H), 2.42–2.47 (m, 2H, CH), 2.59–2.67 (m, 2H, benzyl CH*H_b_* and CH*H_b_’*), 3.78 (s, 6H, CH_3_) 6.72 (d, 2H, J*o* = 8.4 Hz, phenyl C5-H), 6.95 (dd, 2H, J*o* = 8.4 Hz, J*m* = 2 Hz, phenyl C6-H), 7.24 (d, 2H, J*m* = 2 Hz, phenyl C2-H). Anal. Calcd for C_22_H_24_Br_2_O_3_: C, 53.25; H, 4.87; Br, 32.20%. Found: C, 53.27; H, 4.85; Br, 32.18% (Appendix A).

### 4.2. Plant Material and Growth Conditions

Seeds from *Arabidopsis thaliana*, ecotype Columbia, were surface-sterilized using an Ethanol 96%-Bayrochlore 1:10 solution (1 Bayrochlore tab dissolved in 40 mL of water), sown on Petri dishes on a solid medium (halfstrength Murashige and Skoog Sigma salts with vitamins, 0.8% agar, pH 5.7) and kept in the dark at 4 °C for 2 days. The dishes were then transferred in the growth chamber (24 °C/21 °C, 16/8 h day/night, 300 μM/m^−2^s^−1^). As for the treatment with the different inhibitors, two experimental set-ups have been used: for the expression analyses, after stratification, dark-grown 4-day-old seedlings, were transferred on media supplied with increasing concentrations of chemical inhibitors (120, 240 μM), or with an equal volume of its solvent DMSO (Dimethyl sulfoxide), as control, for 2 additional days in the dark. For the phenotypic analyses, after stratification, dark-grown 2-day-old seedlings were transferred on media supplied with increasing concentrations of the inhibitors (60, 120, 240 μM), or with an equal volume of the solvent DMSO as control, for additional 4 days in the dark. The *hac1*-*3* (SALK_080380) and *hac5*-*1* (SALK_074472) single mutant lines are in Col-0 ecotype and have been obtained from the Nottingham Arabidopsis Stock Centre (NASC; https://arabidopsis.info/; accessed on 10 February 2020).

### 4.3. RNA Extraction

Six-day-old dark-grown seedlings have been harvested and immediately frozen in liquid nitrogen. Total RNA was extracted according to Vittorioso et al. (1998) [49]. Briefly, after grinding the tissues in liquid nitrogen, the samples were vortexed for 3 min in the presence of an extraction buffer (0.1 M LiCl, 0.1 M Tris-HCl [pH 8], 0.01 M EDTA, 1% sodium dodecyl sulfate) -phenol-chloroform mixture (1:1:1). Three phenol-chloroform extractions were then performed. RNA was precipitated overnight at 4 °C with 1 volume of 4 M LiCl, followed by a second precipitation with 0.1 volume of sodium acetate.

### 4.4. Expression Analysis

Total RNA was treated with DNase I to remove genomic DNA and 1 µg of it was reverse-transcribed using the PrimeScript^®^ RT reagent kit with gDNA Eraser (TaKaRa; San Jose, CA 95131, USA). Real-time qPCR was performed with SYBR-green I master using the Rotor-Gene Q instrument (Qiagen, Hilden, Germany; https://www.qiagen.com, accessed on 10 February 2020). A total of 1 µL of the diluted cDNA was used, along with the specific primers, listed on Appendix A. Relative expression levels were normalized with the GAPA1 (AT3G26650) and UBQ10 (AT4G05320) reference genes and are presented by the ratio of the corresponding mRNA level of the DMSO-treated sample, which was set to 1. Three independent biological replicates were performed, and one representative experiment is reported with SD values.

### 4.5. Phenotypic Analysis

Seedlings were grown and treated according to the set-up for the phenotypic analysis. The hypocotyls have been measured using IMAGEJ software (Version 1.53t; Biorad; Hercules, CA, USA). The values are the mean of two biological replicates presented with SD values. Significant differences were analyzed by *t*-test (* *p* ≤ 0.01; ** *p* ≤ 0.005).

### 4.6. HPLC Analysis

Four-day-old seedlings were transferred on liquid media (halfstrength Murashige and Skoog Sigma salts with vitamins) supplied with the inhibitor (240 μM), or with an equal volume of its solvent DMSO (Dimethyl sulfoxide), as control, for 2 additional days. Each seedling was washed with water (13 mL) to remove any residual inhibitor on the surface and manually removed from filter paper by a tweezer. The rinse water was transferred to a volumetric flask (50 mL) along with the liquid soil. Methanol (20 mL) was used to rinse the filter paper and the petri dish and transferred to the volumetric flask. Then, the volume was made up to 50 mL with methanol. The control was prepared with the same manner. Next, 10 µL of each solution were injected into the HPLC system consisting of a Shimadzu (Kyoto, Japan) LC-10AD pump, a SIL-10AD autosampler using a 20 µL sample loop, a CTO-10AC column oven, an SPD-10A detector. The detector was set at 220 nm. The analyses were carried by isocratic elution using acetonitrile: water 50:50 v/v as a mobile phase and a C18 column (Waters Symmetry, Milford, MA, USA; 150 mm × 4.6 mm, 3.5 µm) as a stationary phase. A calibration curve (y = 2 × 10^7^x) was constructed with standard solutions of **2c** in acetonitrile at concentrations ranging from 0.0075 to 1.5 mg/mL. Peak areas of **2c** were plotted vs. actual concentrations. Linearity was assessed through evaluation of the coefficient of determination (R^2^ = 0.999).

### 4.7. Evolutive Conservation Analysis

The amino acid sequence of the protein was identified by the code HAC1_ARATH (Q9C5X9). The ConSurf server was used to analyze the conservation of the HAC1 catalytic site (aa 1075–1511). The homologs of the query protein have been searched with the PSI-BLAST algorithm, which searches for sequences homologous to the query through three iterations in the database and calculates the profile using the BLOSUM62 matrix scores (cut-off E-value was set at 0.01), in the “Clean_Uniprot” database. The redundant homologous sequences have been then removed using the CD-HIT clustering method. The 250 sequences closest to the query have been selected, with a minimum identity percentage of 35%.

The resulting sequences have subsequently been aligned using MUSCLE, which uses a progressive alignment strategy followed by three iterative refinements of the multiple alignment obtained. Then, the multiple sequence alignment (MSA) generated was used to reconstruct a phylogenetic tree. Starting from the phylogenetic tree and the MSA, the Rate4Site algorithm calculated the position-specific evolutionary rates of amino acids with an empirical Bayesian method, as it has been shown to significantly improve the accuracy of conservation score estimates compared to the maximum likelihood method. Evolutionary rates were then normalized, grouped, and mapped on the sequence into nine degrees of conservation, 1 to 9, where 1 includes the most rapidly evolving positions and 9 includes the most evolutionarily conserved positions [35].

### 4.8. Expression and Purification of Recombinant HAC1

A 1.31 kb fragment from nt 3241 to 4551 of the *HAC1* coding sequence, corresponding to the HAT catalytic domain (aa 1081–1517), was amplified from a cDNA clone obtained from the Arabidopsis Biological Resource Center (https://abrc.osu.edu/; accessed on 16 July 2021), and cloned NcoI-XhoI in pET28a in frame with a C-terminal 6XHis tag. The recombinant protein was expressed in *E*. *coli* BL21(DE3) cells grown in LB supplemented with kanamycin to OD_600_ 0.5–0.6 and induced with 0.1 mM IPTG at 37 °C for 4 h. Cell pellets from 1-lt cultures were resuspended in 25 mL lysis buffer (MOPS 25 mM pH 7.4, NaCl 300 mM, lysozyme 0.5 mg/mL, Complete protease inhibitor (Roche; Basel, Switzerland), PMSF 1 mM) and sonicated. Triton X-100 was added to 0.5% to the lysate and incubated at room temperature 10 min. After clarification by centrifugation at 21,000× *g* for 20 min, the lysate was chromatographed on HIS-Select (Sigma; Granada, Andalucia, Spain) resin (2 mL) equilibrated in MOPS 25 mM pH 7.4, NaCl 300 mM, imidazole 5 mM. The resin was washed with 25 column volumes of the same buffer followed by 10 column volumes of MOPS/NaCl buffer containing imidazole 15 mM. HAC1 was then eluted at imidazole 100 mM; the protein was concentrated, and buffer exchanged to remove imidazole with Vivaspin 20 10K filters (Merck; Rahway, NJ, USA). Total protein content was determined with the Bradford assay. Enzymatic activity of recombinant HAC1 was assessed with an acetyltransferase in vitro assay (# P-4003 EpiGentek, Farmingdale, NY, USA) according to the manufacturer’s instructions. Briefly, the HAT substrate was captured in strip wells for 45 min. Following washing, strip wells were incubated with acetyl-CoA and partially purified HAC1 catalytic domain, with or without compound **2c** (6 μM), for 45 min at 37 °C. Following washing, wells were incubated sequentially with the capture and detection antibody, followed by addition of the developer solution. Once the solution in the control wells changed to a medium blue, the enzymatic reaction was stopped adding the stop solution. Absorbance was read at λ 450 nm, using UV/Vis microplate spectrophotometer (Thermo Scientific™ Multiskan™ GO; Waltham, MA, USA). Inhibition of HAT activity was measured following the equation: Inhibition % = [1 − (OD (inhibitor sample − blank)/OD (no inhibitor control − blank)] × 100.

### 4.9. Statistical Analysis

Each experiment was performed in duplicate and repeated with two or three independent biological replicates. Results are expressed as mean (except for the expression analysis) ± standard deviation (SD). Two-tailed Student’s *t*-test was used to evaluate statistical significance, respect to the DMSO control (* *p* ≤ 0.01; ** *p* ≤ 0.005).

## Figures and Tables

**Figure 1 ijms-23-10446-f001:**
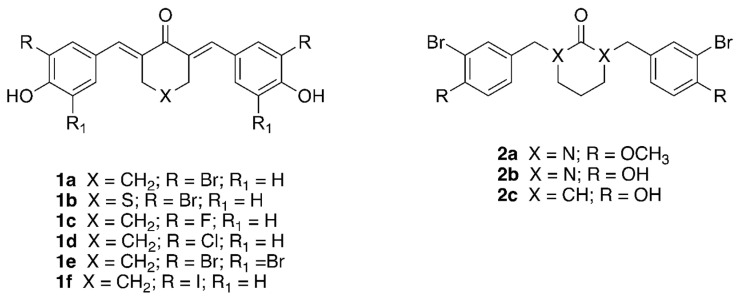
Structures of the cinnamoyl compounds **1a**–**f** and of the saturated derivatives **2a**–**c**.

**Figure 2 ijms-23-10446-f002:**
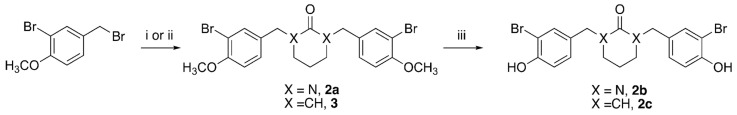
Synthetic route to **2a**–**c** derivatives. i: NaH 60% dispersion in mineral oil, *N*,*N′*-trimethyleneurea, dioxane *dry*, reflux, 24 h, 50% yield; ii: 1. 1-(1-cyclohexen-1-yl)pyrrolidine, DIPEA, CHCl_3_, reflux, 12 h, 2. HCl 1 N, reflux, 4 h, 16% yield; iii: BBr_3_ 1 M in CH_2_Cl_2_, CH_2_Cl_2_ *dry*, −45 °C to rt, 2 h, 35–50% yield (see Section 4).

**Figure 3 ijms-23-10446-f003:**
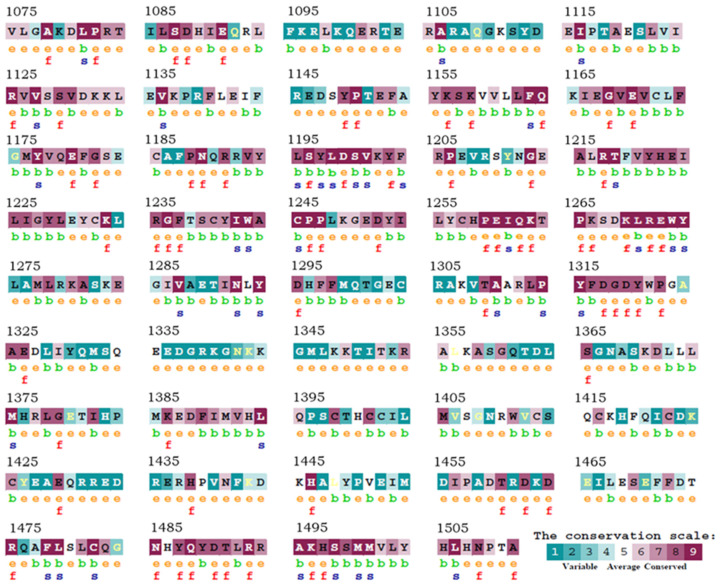
Identification of structurally and functionally conserved amino acid positions. Each residue of the catalytic subunit of Arabidopsis HAC1 is identified by a color representing the degree of evolutionary conservation, cyan to purple, corresponding to variable (grade 1) through conserved (grade 9) positions (first line, uppercase). Below each amino acid is reported the position in the 3D structure of the residue (second line, b and e), and the predicted functionality (third line, f and s). e corresponds to an exposed residue according to the neural network algorithm, b to a buried residue according to the neural network algorithm, f to a predicted functional residue (highly conserved and exposed), s to a predicted structural residue (highly conserved and buried) [35].

**Figure 4 ijms-23-10446-f004:**
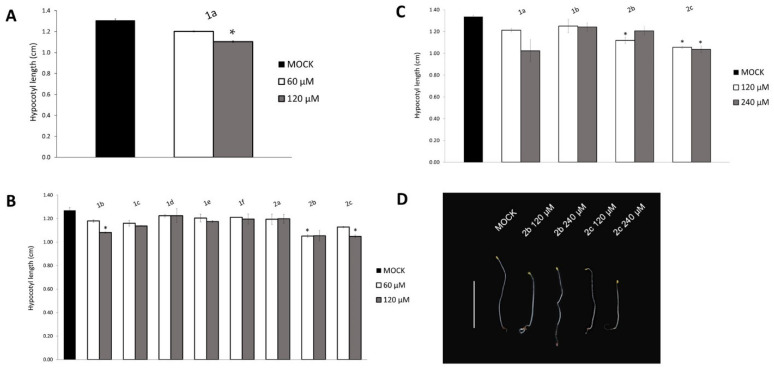
Selection of active compounds through a phenotypic screening. The hypocotyl length of wild type 6-day-old seedlings was treated with: (**A**) **1a** (60, 120 μM), (**B**) **1b**–**f** and **2a**–**c** (60, 120 μM), (**C**) with the selected compounds **1b**, **2b**, and **2c** compared to **1a** (120 and 240 μM). All the samples were compared to the DMSO-treated control (MOCK). (**D**) Six-day-old seedlings treated with DMSO (MOCK) or with compounds **2b** and **2c** (120 and 240 μM). Scale bar, 1 cm. The values of the hypocotyl length are the mean of two biological replicates (50 seedling each) presented with SD values. Significant differences from the mock were analyzed by *t*-test (* *p* ≤ 0.01).

**Figure 5 ijms-23-10446-f005:**
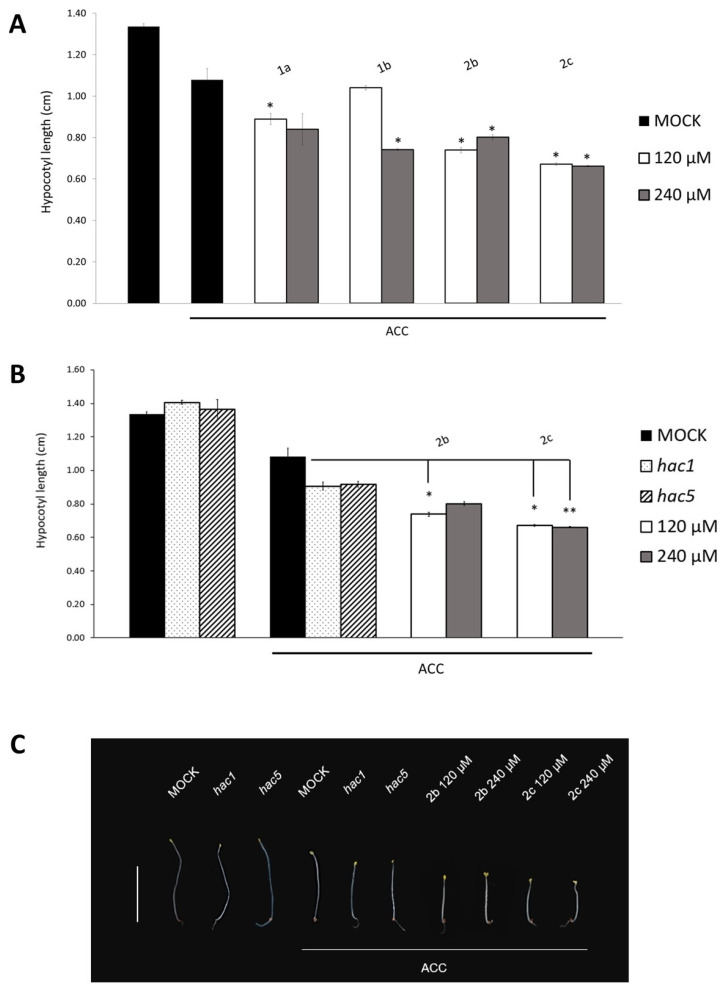
HAC inhibitors result in stronger phenotypes of *hac* mutants, in the presence of ACC. (**A**) The hypocotyl length of wild type 6-day-old seedlings untreated, or treated with the selected compounds **1b**, **2b**, **2c** and **1a** (120 and 240 μM), in the presence of ACC (10 μM), compared to the DMSO-treated control (MOCK). (**B**) The hypocotyl length of untreated (no ACC) or treated (+ACC) wild type, *hac1* and *hac5* single mutants compared to the **2b**- and **2c**-treated wild type (+ACC). (**C**) Six-day-old dark-grown untreated (no ACC) or treated (+ ACC) wild type, *hac1* and *hac5* single mutants compared to the **2b**- and **2c**-treated wild type (+ ACC). Scale bar, 1 cm. The values of the hypocotyl length are the mean of two biological replicates (50 seedling each) presented with SD values. Significant differences from the mock were analyzed by *t*-test (* *p* ≤ 0.01; ** *p* ≤ 0.005).

**Figure 6 ijms-23-10446-f006:**
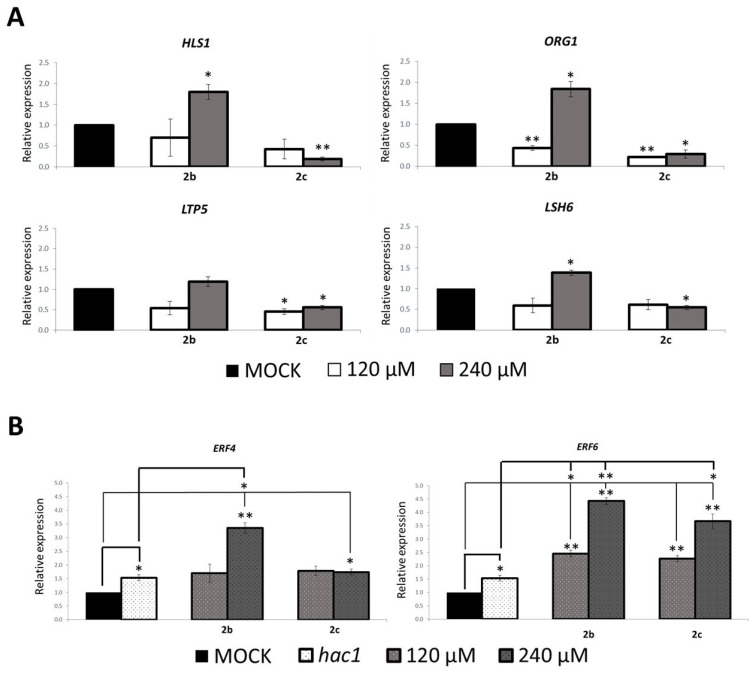
Treatment with **2b** and **2c** affects expression of target genes. Relative expression level of: (**A**) *HLS1*, *ORG1*, *LTP5* and *LSH6* in **2b**- and **2c**-treated 6-day-old dark-grown wild type seedlings, (**B**) *ERF4* (**left**) and *ERF6* (**right**) in **2b**- and **2c**-treated 6-day-old dark-grown *hac1* seedlings. Treated samples were compared to DMSO-treated wild type seedlings (MOCK), which was set to 1. RT-qPCR assays were performed with 1 μL of the diluted cDNA, along with the specific primers, listed in Appendix A. Relative expression levels were normalized with the *GAPA1* (AT3G26650) and *UBQ10* (AT4G05320) reference genes. The values of relative expression level are means of three biological replicates, presented with SD values. Significant differences were analyzed by *t*-test (* *p* ≤ 0.01; ** *p* ≤ 0.005).

**Figure 7 ijms-23-10446-f007:**
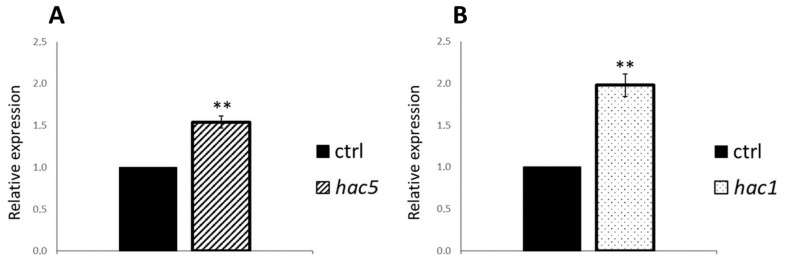
HAC1 and HAC5 mutually regulate their expression. Relative expression level of *HAC1* (**A**) and *HAC5* (**B**) in 6-day-old dark-grown wild type (control, ctrl), *hac5* and *hac1* mutant seedlings, respectively. RT-qPCR assays were performed with 1 μL of the diluted cDNA, along with the specific primers, listed in Appendix A. Relative expression levels were normalized with the *GAPA1* (AT3G26650) and *UBQ10* (AT4G05320) reference genes. The values of relative expression level are means of three biological replicates, presented with SD values. Significant differences were analyzed by *t*-test (** *p* ≤ 0.005).

**Figure 8 ijms-23-10446-f008:**
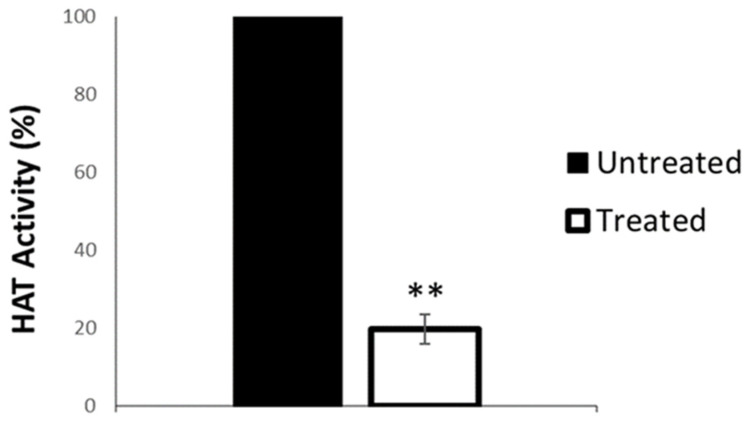
Compound **2c** inhibits acetylase activity of HAC1. HAT activity of partially purified HAC1 catalytic site. Purified protein extracts treated with 6 µM compound **2c** compared to untreated samples. Similar results were obtained from two independent biological replicates, and one representative experiment is presented. The value of HAT activity is the mean of two technical replicates presented with SD values. Significant differences from the untreated samples were analyzed by *t*-test (** *p* ≤ 0.005).

**Table 1 ijms-23-10446-t001:** HPLC analysis of compound **2c** in the medium.

Control	Compound 2c (mg)	Sample	Compound 2c (mg)
C.1	0.91157	S**2c**.1	0.67533
C.2	0.83752	S**2c**.2	0.62548
C.3	0.90223	S**2c**.3	0.59607
average	0.88377		0.63229

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
