# Peer review of "New Inhibitors of the Human p300/CBP Acetyltransferase Are Selectively Active against the Arabidopsis HAC Proteins"

_ijms, 2022, doi:10.3390/ijms231810446_

Round 1

Reviewer 1 Report

The manuscript is of interest. Although a rather simple research, its conclusions are supported by the results. The research design is appropriate and the manuscript is well written. 

I have only minor concerns/suggestions: 

I would recommend the authors to include in their revised manuscript some macroscopic images off treated and untreated plants. Moreover, I would recommend to carefully check the manuscript for some minor linguistic mistakes and some mistakes regarding the format (i.e. unnecessary highlighted text, unnecessary bold, and IJMS format).

Author Response

Point 1:

I would recommend the authors to include in their revised manuscript some macroscopic images off treated and untreated plants. Moreover, I would recommend to carefully check the manuscript for some minor linguistic mistakes and some mistakes regarding the format (i.e. unnecessary highlighted text, unnecessary bold, and IJMS format).

Response 1:

We thank the Reviewer for this suggestion; we have added images of: DMSO-, 2b- and 2c-treated wild type seedlings (Figure 4D), and of untreated (no ACC) or treated (+ ACC) wild type, hac1 and hac5 single mutants compared to the 2b- and 2c-treated wild type (+ ACC) seedlings (Figure 5C). We are sorry for those typos; we corrected the text accordingly.

Reviewer 2 Report

In this publication, new inhibitors are described, which are active against histone acetyltransferases in Arabidopsis thaliana. This publication seems to be within the scope of journal. However it needs several corrections to be more acceptable for publication.

1.      Why was the level of inhibitor absorption by A. thaliana were measured for compound 2c only? Differences in inhibitory activity may also be due to differences in absorption of these compounds by A. thaliana.

2.      Was the metabolism of inhibitors by A. thaliana investigated, in particular what biotransformations these compounds may undergo?

3.      It is absolutely necessary to submit complete spectroscopic documentation in Supplementary Information. Please include the spectra of compound: 2a, 2b, 2c, and 3.

4.      In my opinion, it will be more legible use of word „control” instead of „MOCK”.

5.      Please add information what was the control in the legend to Figure 6.

6.      Lines 4 and 31: please remove color.

7.      Lines 70,79, and 80: names of proteins should be written lowercase, not uppercase.

8.      In Materials and Methods please remove the dot and the space in names of compounds.

9.      All signals should be read from the IR spectrum, not only from the methoxy, hydroxyl or carbonyl groups. Please correct.

10.  Line 597: it should be “mL” instead of “ml”. Please correct evident mistake.

Author Response

Point 1: Why was the level of inhibitor absorption by A. thaliana were measured for compound 2c only? Differences in inhibitory activity may also be due to differences in absorption of these compounds by A. thaliana 

Response 1:

We measured the absorption of coumpound 2c, because, based on our results, it is the most promising compound. Our hypothesis is that compound 2c is more selective than compound 2b for the HAC1 protein. The differences in their inhibitory activity might be due to a compensation phenomenon between HAC1 and HAC5, as we reported in the discussion.

Point 2: Was the metabolism of inhibitors by A. thaliana investigated, in particular what biotransformations these compounds may undergo? 

Response 2:

We thank the Reviewer to raise this point. The pharmacological approach in Arabidopsis thaliana is so far not widespread and, as far as we know, the metabolism of the inhibitors has not been studied. Indeed, this analysis will be topic of our next study. However, we can hypothesize that in plant cells they can occur biotransformations aimed at making these compounds more hydro-soluble.

Point 3: It is absolutely necessary to submit complete spectroscopic documentation in Supplementary Information. Please include the spectra of compound: 2a2b2c, and 3. 

Response 3:

We thank the Reviewer for this suggestion, and we revised the manuscript by adding complete spectroscopic documentation (both IR and NMR spectra) in SI as requested.

Point 4: In my opinion, it will be more legible use of word „control” instead of „MOCK”.

Response 4:

We thank the Reviewer for this suggestion. We used the word mock to mean the DMSO-treated control, as opposed to the untreated control. For lack of ambiguity for the readers, we have indicated throughout the text that MOCK corresponds to DMSO-treated control, while in figure 7 we have named it control (ctrl), as it is not DMSO-treated.

Point 5: Please add information what was the control in the legend to Figure 6.

Response 5:

We are sorry for this oversight; we have edited the legend of figure 6.

Point 6: Lines 4 and 31: please remove color.

Response 6:

We apologize for this formatting error, present only in the pdf format. We have carefully checked the pdf file of the revised manuscript.

Point 7: Lines 70,79, and 80: names of proteins should be written lowercase, not uppercase.

Response 7:

In those lines there are names of genes which, according to the nomenclature in Arabidopsis, must be reported in uppercase italics

Point 8: In Materials and Methods please remove the dot and the space in names of compounds. 

Response 8:

We are sorry for these typos; we corrected the Materials and Methods accordingly.

Point 9: All signals should be read from the IR spectrum, not only from the methoxy, hydroxyl or carbonyl groups. Please correct. 

Response 9:

We thank the Reviewer for this suggestion. Therefore, to improve the comprehensiveness of our manuscript, we add all the IR spectra of the tested compounds 2a-c and 3. Moreover, to increase the straightforwardness for the readers, we have indicated the peaks assignment for the functional groups of each compound’s IR spectrum. Indeed, our main characterization method is the NMR and, given that each NMR spectrum was assigned in full, we added the IR peaks assignments just for the functional groups.

Point 10: Line 597: it should be “mL” instead of “ml”. Please correct evident mistake.

Response 10: 

We are sorry for this mistake; it has been corrected. 

Round 2

Reviewer 2 Report

The manuscript has been carefully revised by the authors and is suitable for publication in its current form.